# good4cir: Generating Detailed Synthetic Captions for Composed Image Retrieval

## Abstract

*Composed image retrieval (CIR) enables users to search images using a reference image combined with textual modifications. Recent advances in vision-language models have improved CIR, but dataset limitations remain a barrier. Existing datasets often rely on simplistic, ambiguous, or insufficient manual annotations, hindering fine-grained retrieval. We introduce good4cir, a structured pipeline leveraging vision-language models to generate high-quality synthetic annotations. Our method involves: (1) extracting fine-grained object descriptions from query images, (2) generating comparable descriptions for target images, and (3) synthesizing textual instructions capturing meaningful transformations between images. This reduces hallucination, enhances modification diversity, and ensures object-level consistency. Applying our method improves existing datasets and enables creating new datasets across diverse domains. Results demonstrate improved retrieval accuracy for CIR models trained on our pipeline-generated datasets. We release our dataset construction framework to support further research in CIR and multi-modal retrieval.*

## 1. Introduction

Composed Image Retrieval (CIR) is an emerging task in vision-language research that allows users to refine image searches by providing both a reference image and a textual modification. While CIR has benefited from advancements in vision-language models (VLMs), the progress of retrieval models remains constrained by limitations in existing datasets. Most CIR datasets are constructed through either manual annotation or automated data mining. Manually labeled datasets, such as CIRR, provide high-quality human descriptions of modifications but are often limited in scale, expensive to create, and prone to inconsistencies in textual annotations. Automatically generated datasets, such as those based on image synthesis or retrieval-based mining, offer scalability but frequently introduce issues such as annotation noise, hallucinated content, or overly simplistic modifications that fail to capture the complexity of real-world retrieval tasks.

In this paper, we introduce a structured framework for generating synthetic text annotations for CIR datasets using

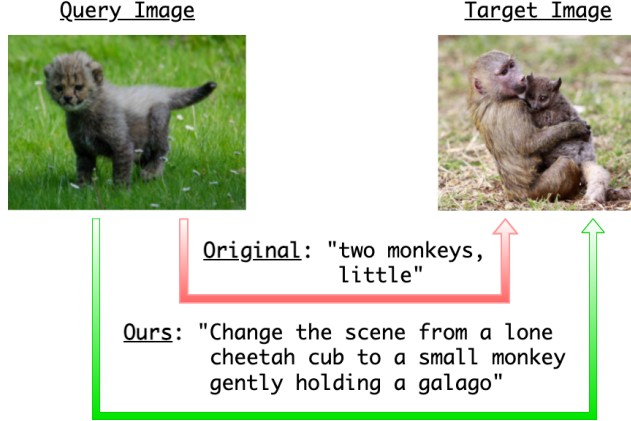

Figure 1. Existing composed image retrieval datasets are costly to construct and often have low quality text annotations. We propose a new approach that leverages VLMs to generate higher quality, synthetic text annotations for composed image retrieval.

a vision-language model-driven pipeline. Our approach consists of three key stages: (1) extracting detailed object-level descriptions from query images, (2) generating a corresponding set of descriptions for target images while ensuring consistency and capturing meaningful differences, and (3) synthesizing natural language modifications that describe the transformations required to reach the target image. This structured approach mitigates common pitfalls in CIR dataset construction, such as hallucinated object descriptions, vague or redundant modifications, and inconsistencies in annotation quality.

We apply our methodology to enhance existing CIR datasets and construct new ones across multiple domains. By evaluating retrieval models trained on datasets generated with our framework, we demonstrate improvements in retrieval accuracy, particularly for fine-grained modifications that require precise object-level reasoning. Our contributions include not only a scalable and effective dataset generation framework but also insights into the impact of dataset composition on CIR model performance. A GitHub link to use our dataset generation pipeline, to access our introduced datasets, and to re-produce our evaluations will be shared in our camera ready submission.

## 2. Related Work

### 2.1. CIR Methods

Modern composed image retrieval (CIR) methods fuse query image and text representations using multimodal vision-language models to retrieve relevant images [4, 5, 9, 20, 27, 31]. Much of the recent work focuses on algorithmic developments to improve CIR performance including through the implementation of attention-based mechanisms [7, 36], denoising [14], and interpolation-based fusion [15]. Generative vision-language models [8, 19] enable training-free CIR, including video-based approaches [2, 28, 30]. Textual inversion techniques [3, 13, 24] learn pseudowords for query images, while other methods refine cross-modal alignments [17, 25, 32, 33] for fine-grained retrieval, particularly in fashion domains.

### 2.2. CIR Datasets

This paper focuses not on algorithmic developments for composed image retrieval (CIR), but on CIR datasets and methods for improving or creating them.

CIR datasets fall into two categories: manually and automatically generated. Manually generated datasets include CIRR [20], derived from NLVR2, which provides human annotations describing image modifications. Although a key benchmark, CIRR has limitations: dependence on NLVR2 image pairs, misaligned captions, and annotations describing only single-object changes [3]. CIRCO [4] addresses these issues by allowing multiple modifications per annotation, sourced from MS-COCO [18], but lacks a training set and serves solely for evaluation.

Automatically generated datasets overcome some of these limitations, leveraging existing labeled data or image-generation tools. Examples include LaSCo [16], synthesizing annotations from large-scale datasets like VQA2.0 [12], and Syn-thTriplets18M [14], generating images via InstructPix2Pix [6]. Domain-specific datasets, such as Birds-to-Words [11] for bird species retrieval and PatternCom [22] for remote sensing, also exist, alongside video retrieval datasets extending CIR temporally [29, 30].

Most relevant to our work is MagicLens [36], which constructs a dataset of 36.7 million triplets using image pairs mined from web pages. After filtering duplicates and low-quality content, captions and instructions are generated via large multimodal and language models. While this methodology is sound and the dataset could be potentially impactful for other researchers working on composed image retrieval, as of March 2025, the dataset is not shared publicly and no code has been shared to replicate it, with the authors stating on GitHub, "We personally would like to release the data but the legal review inside may take years." [1]

Across the CIR datasets that are publicly available, there are a variety of problems, regardless of the method of generation, including queries where the text on its own is sufficient to find

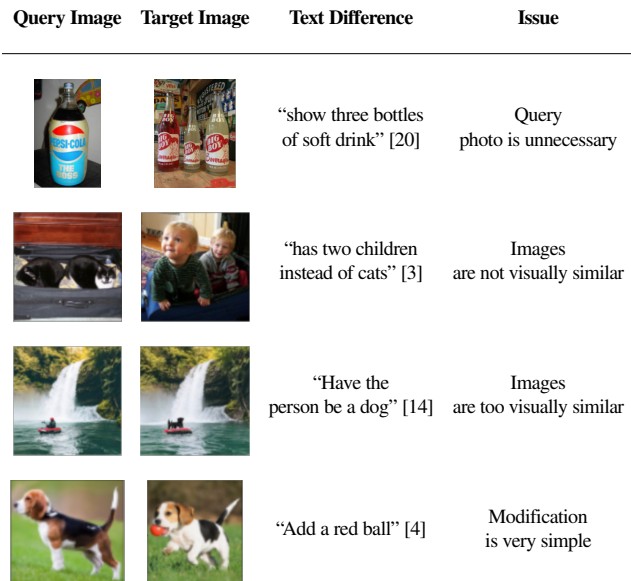

| Query Image | Target Image | Text Difference | Issue |
| --- | --- | --- | --- |
| | | "show three bottles of soft drink" [20] | Query photo is unnecessary |
| | | "has two children instead of cats" [3] | Images are not visually similar |
| | | "Have the person be a dog" [14] | Images are too visually similar |
| | | "Add a red ball" [4] | Modification is very simple |

Figure 2. Qualitative issues with existing CIR datasets.

the target image and issues with the degree of image similarity in the queries. Across existing datasets, the modifications are often overly simple, focusing on a single change to a foreground object. We show examples of these issues in Figure 2. Further, many of the existing CIR datasets such as CIRR and CIRCO are highly general in nature, lacking the specificity required for many domain-specific tasks, such as medical imaging and environmental monitoring. Finally, the scale of many of these datasets is relatively small for any substantial training efforts.

## 3. Method

To improve existing CIR datasets and support the creation of new ones with realistically complex textual modifications, we propose good4cir, a novel pipeline that utilizes a large language model – specifically OpenAI's GPT-4o – to generate CIR triplets. Our approach assumes the presence of a collection of related images, which may originate from an existing CIR dataset with suboptimal annotations or a novel domain containing image pairs (further discussed in Section 3.6). To enhance precision and reduce hallucination, we break down the CIR triplet generation process into focused sub-tasks, designed to encourage the production of fine-grained descriptors [10].

Figure 3 depicts the structure of the proposed synthetic data generation pipeline. good4cir is split into three stages, which we discuss below. In the sections below, we describe the general prompts for each stage. In specific domains, it may be helpful to add additional specification to the prompt, such as the domain of the imagery or type of scene, or a list of objects for the VLM to annotate. We discuss one such case in Section 3.6,

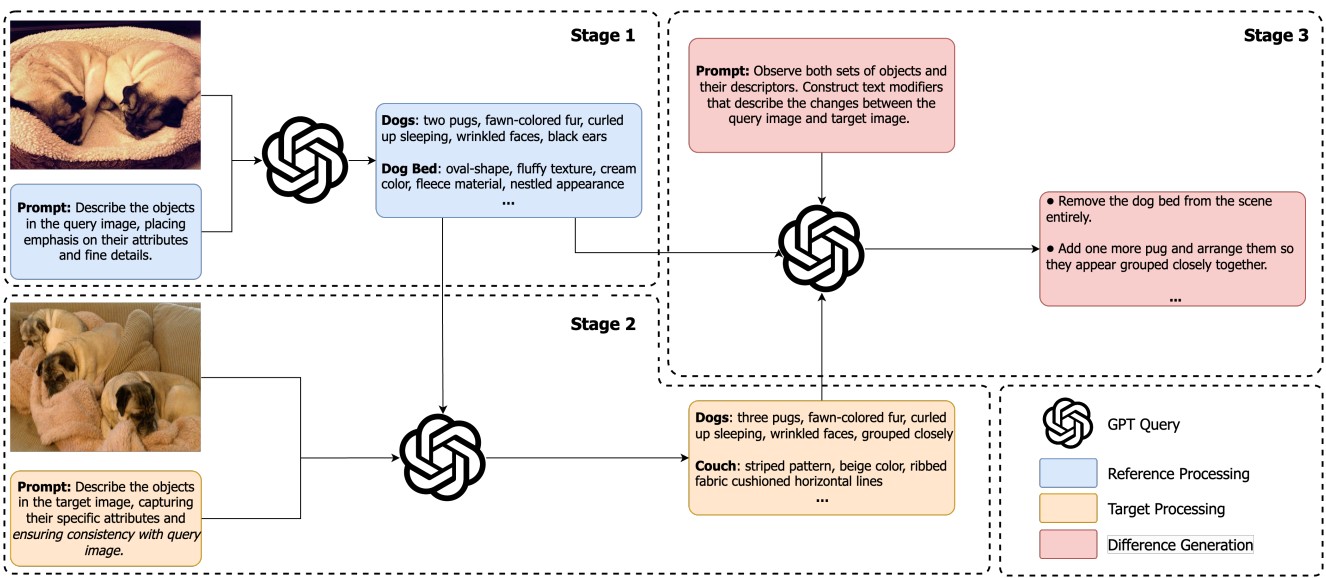

Figure 3. Our synthetic CIR data generation pipeline. The three-stage pipeline uses a structured flow of data to compare a query image and a target image without overwhelming the context window of the VLM to mitigate hallucination. In this figure, the prompts are simplified. The full prompts are discussed in the text.

and include the exact dataset specific prompts in the Appendix. Additionally, in Section 3.5, we demonstrate that this phased approach yields superior CIR triplets when compared with an alternative simpler approach of simply prompting a VLM to describe differences between a pair of images.

### 3.1. Stage 1: Query Image Object Descriptions

In the first stage, the VLM is prompted to generate a list of key objects and descriptors from the query image. Objects are the building blocks of any visually dense image, inherently making them signals of change. Queries used in composed image retrieval reference a specific object and a modifying caption (e.g., *"Find a similar image but change the color of the chair to red"*). By directing the VLM to focus on individual objects, we facilitate a more structured and detailed understanding of image differences.

The general form of the prompt for this stage is:

*"Curate a list of up to X objects in the image from most prominent to least prominent. For each object, generate a list of descriptors. The descriptors should describe the exact appearance of the object, mentioning any fine-grained details.*

*Example: Object Name: ["object description 1", "object description 2", . . . , "object description N"]*

*Format objects and descriptors as a JSON output."*

The example should be constructed for the specific domain, and the quantity for $X$ can be modified depending on the density of objects in the dataset and desired number of outputs.

### 3.2. Stage 2: Target Image Object Descriptions

In the second stage, the VLM is prompted to derive a similar list from the target image by comparing it against the list of objects from the query image, ensuring consistency and making modifications when necessary. This is done by passing both the following prompt and the output from the first stage into the VLM:

*"Here is an image and a list of descriptors that describe a different image. Curate a similar list for this image by doing the following:*

*1. If there is a new object in this image that isn't described in the description of the other image, generate a new set of descriptors.*
*2. If the description of an object from the other image matches the appearance of an object in this image, use the exact same list of descriptors.*
*3. If the object appears different in this image in comparison to the description from the other image, generate a new set of descriptors.*

*Format objects and descriptors as a JSON output."*

### 3.3. Stage 3: Describing Differences

In the final stage, the text outputs from the first two stages are passed into the VLM with the following prompt:

*"The following are two sets of objects with descriptors that describe two different images that have been determined to be different in some ways. Analyze both lists and generate*

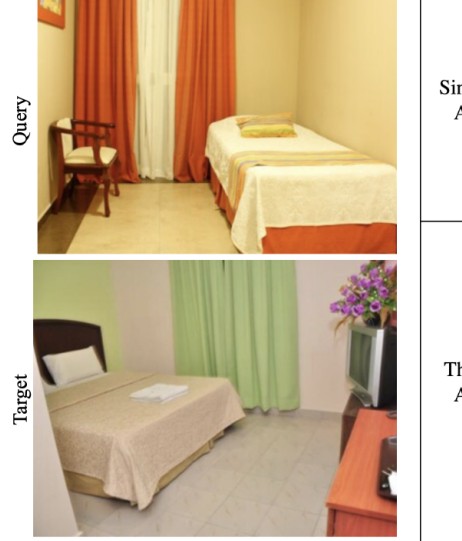

| | | |
|---|---|---|
| Query | | **Single-Stage Approach** | • Swap the orange curtains for green ones to create a fresher and more calming atmosphere.
• Replace the orange bed skirt with a beige or light brown one to match a more neutral color scheme.
• **Change the chair to a darker wooden style with a more cushioned seat for a cozier touch.**
• Update the bedding with a subtle, patterned beige cover to create a more sophisticated and uniform look.
• Replace the artwork for a simple decorative vase with fresh flowers to add color and vibrancy to the room.
• Adjust the lighting to be softer and less yellow-toned to create a more inviting and relaxing ambiance.
• Replace the tiled flooring with a lighter, more uniform shade to brighten up the space.
• **Introduce a small wooden nightstand beside the bed to provide a functional and aesthetically balanced layout.** |
| Target | **Three-Stage Approach** | • Change the bed from single size to full size.
• Raise the height of the bed from low to medium height.
• Replace the decorative pillows with a single, rectangular, white pillow.
• Change the decorative pillows to have a plush appearance with soft texture.
• Replace the white blanket with a beige one.
• Ensure the blanket maintains a thin layer, neat and tidy appearance, and smooth texture.
• Replace the rust-colored bedskirt with a light brown bedskirt.
• Change the bedskirt design from ruffled to plain.
• Swap the rust-colored, dual-layer curtains for light green, single-layer curtains.
• Introduce a headboard with a dark brown color, wooden material, and curved top.
• Add a CRT model television, black in color and positioned on a stand.
• Introduce a rectangular wooden table to support the TV.
• Add decorative artwork of flowers with purple and pink colors, green leaves, arranged in a bouquet in a vase on the table.
• Change the flooring color from light beige to light grey. |

Figure 4. Comparing the direct single-stage prompting method for capturing differences, versus using good4cir's three-stage approach.

*short and comprehensive instructions on how to modify the first image to look more like the second image. Be sure to mention what objects have been added, removed, or modified. Don't mention "Image 1" and "Image 2" or any similar phrasing. Focus on having variety in the styles of captions that are generated, and make sure they mimic human-like syntactical structure and diction."*

good4cir's three-stage pipeine is aimed at addressing two fundamental issues that arise when working with VLMs:

1. **Hallucination:** VLMs generate captions that describe objects or attributes that are not actually present in the image. The multi-stage pipeline mitigates this by guiding the model to focus on concrete objects, rather than deriving a wholistic interpretation of the scene that may introduce imaginary objects or features.
2. **Limitations in Fine-Grained Captioning:** VLMs are proficient in generating relatively descriptive captions but may lack the granularity demanded by fine-grained retrieval tasks. A single-stage, direct captioning approach may lead to a vague or uninformative understanding of the object's appearance. This idea motivates the three-stage procedure.

### 3.4. Stage 4: Caption Permutations

After running the first three stages, we have a dataset that consists of a number of image pairs and synthetically generated text captions describing specific differences between the images. In order to construct captions that contain more complex text differences, we implemented an automated procedure to combine individual captions into compound sentences.

For exactly two captions, we joined them by removing the period from the first caption, adding a comma and the word 'and', and converting the second caption's initial character to lowercase, resulting in a natural-sounding compound sentence. For combinations involving three captions, we sequentially combined the first two captions with commas, ensuring all intermediate captions began with lowercase letters, and added the conjunction 'and' before the final caption. The final datasets include each original caption on its own, and then randomly sampled combinations of up to three captions, ensuring no caption was used more than once within compound sentences. Captions containing the verbs 'maintain' or 'ensure' were excluded, as they do not indicate actual differences between the query and target images.

We then utilized the CLIP tokenizer from OpenAI's CLIP-ViT model (base-patch32) to validate each generated caption, discarding combinations exceeding the tokenizer's 77-token limit. Combination generation continued until either all available sentences were exhausted or a predefined limit of 60 total combined sentences per image pair was reached.

### 3.5. Comparison to a Single-Stage Approach

An alternative to good4cir's three-stage approach would be a single-stage approach, where the VLM is directly prompted to describe the differences between a pair of images. For comparison, we consider the following prompt:

*"The following are two different rooms that have been determined to be different in some ways. Analyze both lists and generate short instructions on how to modify the first image to look more like the second image. Don't mention "room 1" and "room 2" or any similar phrasing. One caption should discuss one modification that needs to be made to one*

| Dataset | Train | | | Val | | | Test | | | Average Metrics | |
| | Image Pairs | CIR Triplets | Total Images | Image Pairs | CIR Triplets | Total Images (w/ Distractors) | Image Pairs | CIR Triplets | Total Images (w/ Distractors) | Avg. Prompt Tokens | Avg. Output Tokens |
|---|---|---|---|---|---|---|---|---|---|---|---|
| $CIRR_R$ | 28,225 | 199,350 | 16,939 | 4,184 | 22,620 | 2,297 | – | – | – | 1,600 | 670 |
| Hotel-CIR | 65,364 | 415,447 | 129,225 | 2,092 | 13,298 | 14,549 | 2,069 | 13,178 | 14,404 | 3,310 | 1,750 |

Table 1. Dataset Statistics

*element of the room. If one object has multiple modifications that need to be made, include each modification in a separate caption. Make sure to focus on having variety in the styles of captions that are generated, and make sure they mimic human-like conversational syntactical structure and diction."*

Figure 4 compares the output of the single-stage, end-to-end approach with that of the good4cir pipeline. In the captions generated by direct captioning method, a modification to a chair in the room is described, but no chair exists in the target image. Similarly, the VLM incorrectly describes the addition of a nightstand in the second image, despite there being no nightstand. Both errors emphasize the hallucination issue with VLMs as well as their tendency to confuse objects and ideas when operating in an enlarged context window. Additionally, in the first set of captions, the model simply mentions the addition of a flower, whereas the second set provides details on the exact colors of the flowers and leaves, as well as their arrangement. This level of granularity is achieved through the structured pipeline, demonstrating the limitations of direct captioning.

### 3.6. Constructing New CIR Datasets

CIR datasets consist of triplets of query images, target images, and the text that describes the modification between the two. Many CIR datasets also include distractor images that are similar to the query, but do not necessarily match the text modification. Our proposed method for generating CIR captions assumes that the query-target image pairs already exist, as in the case of rewriting the captions for existing CIR datasets.

It is also possible to construct new CIR datasets by mining image pairs in existing image datasets that are visually similar but likely to contain differences. This is a property that is especially likely to be found in fine-grained domains, where there are large numbers of visually similar images from different classes. To mine CIR pairs from fine-grained domains, we use a combination of two different image representations:

1. **Learned Image Embedding:** Using either a domain-specific embedding model (i.e., one trained on a specific dataset) or a general-purpose model such as CLIP's image encoder, we can identify the most semantically similar image for each image in a dataset. This process generates pairs of related images based on the similarity notion that was optimized over during the model training.
2. **Perceptual Hashing:** We use perceptual hashing and select both a minimum and maximum hash distance, allowing

us to identify pairs that structurally and visually similar, without being identical.

The exact similarity thresholds, and relative importance of the learned image similarity and perceptual hash similarity vary as a function of the dataset.

## 4. Datasets

We use our proposed approach to generate synthetic text annotations for two new datasets – $CIRR_R$, which is a re-written version of the CIRR dataset, and Hotel-CIR, a new CIR dataset focused on hotel recognition, a very object-centric fine-grained problem domain. Table 1 includes details on the number of image pairs, generated CIR triplets and total images (including distractors) in the training, validation and test sets, as well as the average number of GPT-4o tokens used per prompt.

### 4.1. $CIRR_R$

We use our approach to re-write the captions for the CIRR training and validation sets. As of March 2025, using the gpt-4o model and the OpenAI Batch API, it cost just about $200 to generate all of the synthetic captions for $CIRR_R$.

Figure 5 (top) shows several examples of image pairs from the original CIRR dataset with the original CIRR text difference caption, and a sample of our re-written captions. These examples show that not only does our proposed approach generate many text prompts for each image query, but those prompts are also significantly richer in both the variations they describe and the language and grammar that they use to describe them. Additional examples can be found in the Appendix.

The CIRR test set is not publicly shared. This limits the relevance of our re-written captions for evaluating performance on the CIRR test set, as those captions are still in the same style as the original training set – however, in Section 6 we show that training on the rewritten dataset yields performance improvement on the zero-shot CIR dataset CIRCO.

### 4.2. Hotel-CIR

In order to construct the Hotel-CIR dataset, we start from the Hotels-50K dataset [26]. The hotels domain is ideal for this pipeline because the scenes in the images are dense in terms of the number of objects in any given image, and there are large numbers of visually similar images, requiring CIR models to learn subtle visual differences and rich representations of textual and semantic features.

| Query Image | Target Image | Text Modifiers |
|---|---|---|
| 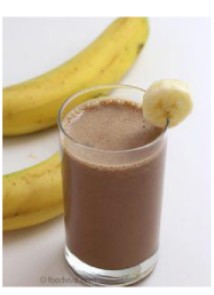 | 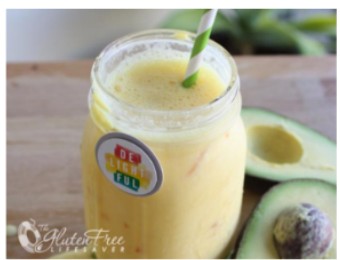
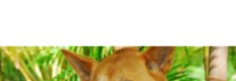 | **CIRR**:
• have an avocado in the background

**CIRR_R**:
• Remove the glass of chocolate smoothie with banana slice garnish
• Replace the banana with an avocado sliced in half, showing light green flesh and a large seed
• Change the chocolate smoothie to a yellow smoothie with a creamy texture, pale yellow color, and tiny bubbles on top
• Swap the transparent glass for a mason jar with an embossed logo, open top, and add a green and white spiral-patterned straw
• Add a circular sticker label with multicolored text reading 'DE LIGHT FUL' affixed near the top
• Introduce a flat, light brown, smooth wooden surface with natural grain lines as the background |
| 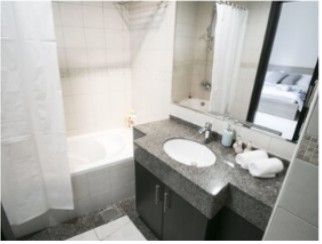 | 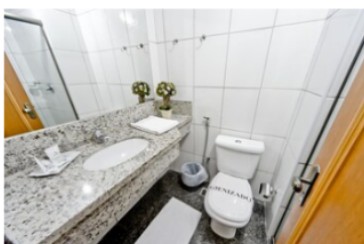 | **CIRR**:
• instead of rabbits dog is sitting in grass

**CIRR_R**:
• Remove both guinea pigs from the scene
• Introduce a tan and white short-haired dog with alert ears and a black nose in a reclining position
• Add a bright green fern with feathery leaves behind the dog to enrich the backdrop
• Include a cylindrical tree trunk with brown bark beside the dog to enhance the natural setting
• Alter the vivid green grass to appear in tandem with the new elements, supporting a cohesive natural environment |
| 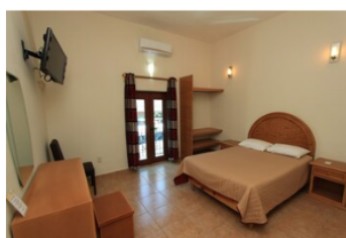 | 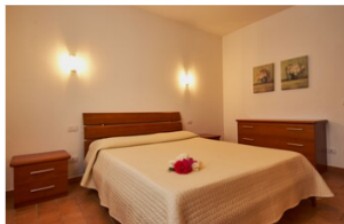 | **Hotel-CIR**:
• Remove the bathtub, Add a toilet with a white porcelain material, standard size, compact shape, attached tank, and chrome handle
• Place a white rectangular bath mat with a soft texture and non-slip backing on the floor
• Add decorative plants with green leaves in small white pots and place them on the countertop
• Install a wooden door with light brown color, modern handle, and smooth surface
• Ensure the door has a hinged design
• Ensure the decorative plants are artificial and neatly arranged
• Ensure the added door has a clean appearance and light finish
• Remove the shower curtain |
| | | **Hotel-CIR**:
• Replace the tan-colored blanket with a beige-tan blanket on the bed, Remove the two pillows from the bed
• Change the nightstand to have a medium brown color instead of tan
• Modify the nightstand to have two drawers instead of a single drawer,
• Replace the simple headboard of the bed with a headboard that has medium-height horizontal slats and a smooth texture
• Add a wooden table with three drawers, matching the nightstand and bed frame, positioned against the wall
• Include two pieces of floral artwork with green and beige coloring, framed and mounted on the wall side by side above the wooden table
• Install two mounted lights on the walls
• Ensure no bedskirt is visible around the bed
• Make sure that the bedspread on the bed is centered |

Figure 5. Example generated text differences for the CIRR_R (top) and Hotel-CIR (bottom) using our synthetic data generation pipeline. For CIRR_R, we include the original caption as well.

We construct (query, target) image pairs by first computing image embeddings for the images in the Hotels-50K dataset using the pre-trained model from [34], and selecting the nearest neighbor for each image that is not from the same hotel (to guarantee that there are possible modifications to describe in text). We then use perceptual hashing to filter image pairs that are either too dissimilar or nearly identical, using a similarity threshold between 25 and 35 (inclusive). Near identical matches can occur in the original Hotels-50K dataset, as different hotels in the same chain occasionally use the same promotional images. Combining the learned image similarity and the perceptual hashing thresholding yields a set of image pairs that can be passed through the synthetic data pipeline to generate data triplets of a CIR dataset.

The specific prompts used at each stage of the pipeline to generate the Hotel-CIR dataset can be found in the Appendix. These captions are slightly modified from the "general" case, as including domain-specific information (such as the fact that

these images come from hotel rooms, and providing a list of specific objects of interest) yields improved text differences.

We additionally include distractor images in the Hotel-CIR dataset. To find reasonable distractor images, we embed the entire Hotels-50K dataset using OpenAI's CLIP-ViT image encoder (base-patch32). For every (query, target) pair in the proposed dataset, we find any other images that have higher cosine similarity in the CLIP image embedding space than the query and target. We randomly sample up to 5 of these images as distractors for every image pair in CIR. The same image may be a distractor for multiple pairs. These distractors ensure that the composed image retrieval task in this dataset is challenging, and that models trained on it must actually learn to incorporate the information from the text difference caption, rather than simply finding visually similar image pairs.

Figure 5 (bottom) shows several examples of CIR triplets from this new dataset, and additional examples can be found in the Appendix.

## 5. Evaluation

To demonstrate how effective our proposed pipeline is at generating high-quality data, we conduct a series of experiments training simple CIR models on both existing datasets and our synthesized datasets created using the good4cir approach. We train supervised models based on the CLIP [23] ViT-B backbone. We train three modules: an image encoder $f_I$, a text encoder $f_T$, and a multimodal fusion mechanism $f_F$, where $f_I, f_T$ are the CLIP image and text ViT-B models, respectively. $f_F$ is implemented using 4 sequential cross attention layers using the text tokens as $Q$ and the image tokens and previous outputs as $KV$, followed by an attentional pooling as defined by Yu et al. [35]. We define a forward pass through the entire model as $f(Q,M) = f_F(f_I(Q), f_T(M))$ for a query image and modification text pair $Q, M$. This model is optimized contrastively with the following loss function, given a batch of size $N$, $\{(Q_i, M_i, T_i), i \in \{1,2,...,N\}\}$:

$$\mathcal{L} = \frac{\exp(\text{sim}(f(Q_i, M_i), f_I(T_i)) / \tau)}{\sum_{j=1}^{N} \exp(\text{sim}(f(Q_j, M_j), f_I(T_j)) / \tau)}$$

This framework is optimized with AdamW [21] with a weight decay of `1e-2`.

We trained this model on the following datasets and their combinations:

1. CIRR (baseline): Composed Image Retrieval on Real-life images dataset.
2. CIRR$_R$: a variant of the CIRR dataset rewritten using the proposed pipeline.
3. Hotel-CIR: a composed image retrieval dataset generated for the hotels domain using the VLM-powered pipeline.

Because the good4cir pipeline generates a number of captions for every (query, target) image pair, the CIRR$_R$ dataset includes a significantly larger number of triplets than the original

| Method | R@1 | R@2 | R@5 | R@10 | R@50 |
|---|---|---|---|---|---|
| CIRR | 16.506 | 25.205 | 41.181 | 56.289 | 82.072 |
| CIRR$_R$ | 9.470 | 16.337 | 29.759 | 43.398 | 72.265 |
| CIRR + CIRR$_R$ | **19.181** | **29.976** | **47.566** | **61.157** | **86.048** |

Table 2. Evaluation on CIRR test set. We evaluate CIRR$_R$ and Hotel-CIR against CIRR (baseline) using a performance metric of Recall@K (or R@K). The best results are bolded.

CIRR dataset. To ensure fairness in our evaluation, when we train on CIRR$_R$, we sample the synthetic captions and only include a single caption for each image pair. It likely would be beneficial to train on the full dataset, but that would make the comparison between models trained on CIRR and CIRR$_R$ unfair.

## 6. Results

To evaluate the quality of the data produced by the good4cir pipeline, we compare retrieval performance across various training setups: (1) models trained on existing CIR datasets (CIRR), (2) models trained on good4cir generated datasets (CIRR$_R$, Hotel-CIR), and (3) models trained on a combination of both dataset types. All model setups were evaluated on the Hotel-CIR, CIRR, and CIRCO test sets. The results from these experiments are summarized in Tables 2, 3, and 4.

### 6.1. CIRR Evaluation

Table 2 summarizes the results from training on the CIRR, CIRR$_R$, and their aggregate datasets and evaluating on the original CIRR test set. Training with only CIRR$_R$ captions degrades retrieval performance compared to training on the original CIRR training set. Since the text modifiers in the CIRR$_R$ dataset were reformulated to introduce greater semantic complexity, they are no longer well aligned with the query composition of the CIRR test set. Consequently, the model struggles to align text queries to their corresponding images. However, when CIRR and CIRR$_R$ are combined, the model exceeds that of the CIRR baseline, suggesting that the diverse captioning offered by the CIRR$_R$ strengthens the model's ability to generalize when integrated with CIRR.

### 6.2. Hotel-CIR Evaluation

The model trained only on the original CIRR captions, and evaluated on the Hotel-CIR test set achieves the lowest recall scores across all thresholds, signifying its limitations in fine-grained composed image retrieval tasks. By comparison, training on only CIRR$_R$ data offers a small boost in performance which is most apparent at higher recall levels. However, the retrieval accuracy achieved when coupling these datasets together surpasses that of any one dataset alone. It is reasonable to assume that the model benefits from the greater diversity in length, complexity, and style of training examples provided by the combined training set.

| Method | R@5 | R@10 | R@50 | R@100 |
|---|---|---|---|---|
| CIRR | 1.27 | 2.03 | 5.80 | 9.07 |
| CIRR$_R$ | 1.61 | 2.75 | 7.52 | 11.22 |
| CIRR + CIRR$_R$ | 2.07 | 3.20 | 8.66 | 13.09 |
| Hotel-CIR | 8.32 | **12.35** | **26.07** | **34.41** |
| CIRR + Hotel-CIR | 7.85 | 11.77 | 24.72 | 32.70 |
| CIRR$_R$ + Hotel-CIR | **8.62** | 12.23 | 25.73 | 34.15 |
| CIRR + CIRR$_R$ + Hotel-CIR | 8.57 | 12.20 | 25.69 | 34.04 |

Table 3. Evaluation on Hotel-CIR test set. We evaluate training on CIRR (baseline), CIRR$_R$ and Hotel-CIR using the performance metric of Recall@K. The best results are bolded.

Still, training exclusively on Hotel-CIR data yields the greatest performance boost. Given that it is a domain-specific dataset that places an emphasis on small, object-level modifications, Hotel-CIR better guides the model in understanding subtle visual differences. As shown in Table 3, Hotel-CIR achieves the highest recall accuracies at R@10, R@50, R@100, and third highest at R@5. This is likely due to the CIRR$_R$ introducing specific concepts that help retrieval in a few select cases. Otherwise, coupling the Hotel-CIR dataset with any data set from the CIRR domain (CIRR or CIRR$_R$) negatively impacts retrieval performance. Since the concepts of the CIRR domain have minimal overlap with the hotels domain, they likely disrupt the patterns that the model is trying to learn from hotel-related images, introducing noise into the model.

### 6.3. CIRCO Evaluation

CIRCO is a zero shot composed image retrieval dataset that has multiple possible targets per query. In comparison to CIRR, the CIRCO captions are generally longer and more descriptive in their composition, making its test set a more relevant evaluation for the utility of the good4cir-generated datasets than the original CIRR test set.

Table 4 shows results on the CIRCO test set when training with CIRR, CIRR$_R$ and their combinations, as well as combining them with the Hotel-CIR dataset for a single more expansive training dataset. Training on the CIRR$_R$ dataset exceeds the performance of training only on CIRR, and combining them together achieves slightly better performance still. This indicates that CIRR$_R$ is better aligned with the textual structure and complexities of the CIRCO test set than CIRR. We further demonstrate this by training on the aggregate of CIRR, CIRR$_R$, and Hotel-CIR which nearly doubles the mAP score at mAP@5, mAP@10, mAP@50, and mAP@100. These results suggest that the captions generated by the good4cir pipeline improve the model's ability to generalize across different retrieval tasks of varying complexities.

### 7. Limitations

While the proposed approach to generating synthetic text annotations for CIR datasets mitigates known limitations of

| Method | mAP@5 | mAP@10 | mAP@25 | mAP@50 |
|---|---|---|---|---|
| CIRR | 2.54 | 2.78 | 3.14 | 3.54 |
| CIRR$_R$ | 2.72 | 3.29 | 3.84 | 4.12 |
| CIRR + CIRR$_R$ | 2.84 | 3.43 | 4.21 | 4.60 |
| CIRR + CIRR$_R$ + Hotel-CIR | **4.64** | **5.39** | **6.38** | **7.04** |

Table 4. Evaluation on CIRCO test set. We evaluate training on CIRR (baseline), CIRR$_R$ and Hotel-CIR using the performance metric of mAP@K. The best results are bolded.

VLMs, several challenges persist:

- **Hallucination:** The three-stage pipeline reduces but does not fully eliminate hallucination. Particularly when query and target images are highly similar, the VLM occasionally describes objects not present in either image. Hallucinations are less frequent in datasets with more visually distinct image pairs (e.g., CIRR dataset).
- **Counting:** VLMs often inaccurately count objects, resulting in captions that correctly identify objects but incorrectly specify their quantity.
- **Sentence Structure:** Despite prompts requesting varied styles, chat-based LLM outputs often exhibit limited stylistic diversity. Future work could address this by adding a post-processing step to rewrite captions in diverse styles.
- **Object-centric Focus:** The pipeline primarily captures variations in individual objects, limiting its effectiveness for non-object-centric datasets and abstract, conceptual differences. For instance, it might describe furniture changes in a room but miss broader shifts, such as from a modern to a traditional ambiance.
- **Cost:** The proposed method relies on OpenAI's GPT-4o, incurring a per-query cost. While substantially cheaper than human annotation, this expense remains noteworthy. We explored open-source VLM alternatives but found GPT-4o significantly superior.

### 8. Conclusion

In this work, we presented good4cir, a structured and scalable pipeline for generating synthetic, high-quality text annotations for Composed Image Retrieval datasets. By leveraging advanced vision-language models and a carefully designed multi-stage prompting strategy, our approach generates richer and more diverse textual annotations than existing datasets. We introduced two new datasets, CIRR and Hotel-CIR, created using good4cir, and demonstrated through evaluations on composed image retrieval benchmarks that training with these datasets improves composed image retrieval accuracy in general. Our datasets and publicly available construction framework, which can be found at `https://github.com/tbd/after/camera/ready` aim to facilitate further progress and innovation in composed image retrieval and broader multimodal retrieval research.

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
