# good4cir: Generating Detailed Synthetic Captions for Composed Image Retrieval

## Appendix

### 1. Synthetic Dataset Specific Prompts

In this section, we provide the exact prompts used for each stage to generate the synthetic text annotations for the $CIRR_R$ and Hotel-CIR datasets.

### 1.1. $CIRR_R$

We curated the prompts below to be used in the rewriting of the existing CIRR dataset from the original (image query, reference text, target image) triples.

*Stage 1:* *"Curate a list of up to 6 defining objects from most prominent to least prominent. For each object, generate a list of at least 4-6 descriptors. The descriptors should describe the exact appearance of the object, mentioning fine-grained details.*

*Example: Picture Frame : ["rectangular shape", "black thin frame", "black and white photo", "mounted on wall", "architectural content", "traditional style"].*

*Format objects and descriptors as a JSON output."*

*Stage 2:* *"Here is an image and a list of descriptors that describe a different image. Curate a similar list for this image by doing the following:*

*1. If there is a new object in this image that isn't described in the description of the other image, generate a new set of descriptors.*
*2. If the description of an object from the other image matches the appearance of an object in this image, use the exact same list of descriptors.*
*3. If the object appears different in this image in comparison to the description from the other image, generate a new set of descriptors.*

*Stage 3:* *"The following are two sets of objects with descriptors that describe two different images that have been determined to be different in some ways. Analyze both lists and generate 3-5 short and comprehensive instructions on how to modify the first image to look more like the second image. Be sure to mention what objects have been added, removed, or modified. Don't mention "Image 1" and "Image 2" or any similar phrasing. Focus on having variety in the styles of captions that are generated, and make sure they mimic human-like syntactical structure and diction."*

### 1.2. Hotel-CIR

For the Hotel-CIR dataset, we use the following prompts at each stage of the pipeline:

*Stage 1:* *"Curate a list of up to 10 defining objects in the image of the room from most prominent to least prominent.*

*Choose the most appropriate label from the following list to name the object: Bed, Pillow, Decorative Pillow, Blanket, Bed skirt, Headboard, Footboard, Runner, Nightstand, Floor Lamp, Bedside Lamp, Television, Window, Curtains, Couch, Ottoman, Chair, Desk, Table, Cabinet, Shelf, Artwork, Walls, Wallpaper, Flooring, Moldings, Engravings, Mirror, Bathroom Towels, Bath Mat, Hair Dryer, Shower Head, Shower Curtain, Sink, Counter Top, Toilet, Waste Basket, Bathtub, Vanity Mirror.*

*If there is an object that cannot be categorized into one of these labels, assign a label as you see fit. For each object, generate a list of as many descriptors as possible, at least 8-10. The descriptors should describe the exact appearance of the object, mentioning fine-grained details.*

*Example: Picture Frame : ["rectangular shape", "black thin frame", "black and white photo", "mounted on wall", "architectural content", "traditional style", "vertical orientation", "smooth texture", "no visible glass reflection", "minimalistic design"]*

*Format objects and descriptors as a JSON output."*

*Stage 2:* *"Here is an image of a room and a list of descriptors that describe a different room. Curate a similar list for this room by doing the following:*

*1. If the description of an object from the other room matches the appearance of an object in this room, use the exact same list of descriptors.*
*2. If the object appears different in this picture in comparison to the description from the other room, generate a new list of descriptors.*
*3. If there is a new object in this image that isn't described in the description of the other room, generate a new set of descriptors.*

*Choose the most appropriate label from the following list to name the new objects: Bed, Pillow, Decorative Pillow, Blanket, Bed skirt, Headboard, Footboard, Runner, Nightstand, Floor Lamp, Bedside Lamp, Television, Window, Curtains, Couch, Ottoman, Chair, Desk, Table, Cabinet, Shelf, Artwork, Walls, Wallpaper, Flooring, Moldings, Engravings, Mirror, Bathroom Towels, Bath Mat, Hair Dryer, Shower Head, Shower Curtain, Sink, Counter Top, Toilet, Waste Basket, Bathtub, Vanity Mirror.*

*If there is an object that cannot be categorized into one of these labels, assign a label as you see fit. For each differing object in the room, generate a list of as many*

*descriptors as possible, at least 8-10. The descriptors should describe the exact appearance of the object, mention fine-grained details.*

*Example: Picture Frame : ["rectangular shape", "black thin frame", "black and white photo", "mounted on wall", "architectural content", "traditional style", "vertical orientation", "smooth texture", "no visible glass reflection", "minimalistic design"]*

*Format all output as a JSON similar to the other room's descriptors."*

***Stage 3:** "The following are two sets of objects with descriptors that describe two different rooms that have been determined to be different in some ways. Analyze both lists and generate short instructions on how to modify the first image to look more like the second image. Don't mention "room 1" and "room 2" or any similar phrasing. One caption should discuss one modification that needs to be made to one element of the room. If one object has multiple modifications that need to be made, include each modification in a separate caption. Make sure to focus on having variety in the styles of captions that are generated, and make sure they mimic human-like conversational syntactical structure and diction. Generate at least 8 difference captions; however, the goal is to generate as many as possible."*

## References

[1] Zheyuan Liu, Cristian Rodriguez-Opazo, Damien Teney, and Stephen Gould. Image retrieval on real-life images with pre-trained vision-and-language models. In *Proceedings of the IEEE/CVF International Conference on Computer Vision (ICCV)*, pages 2125–2134, 2021. 3

| Query Image | Target Image | Text Modifiers |
|---|---|---|

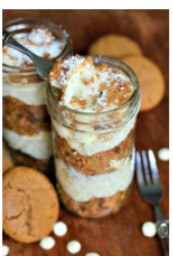
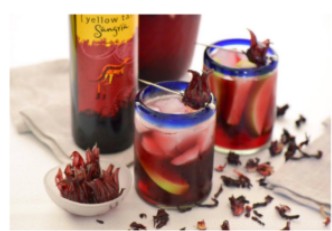

**CIRR**:
· has fruit and the drink is pink

**CIRR$_R$**:
· Remove the mason jar and dessert contents.
· Replace with a tall sangria bottle featuring a yellow label and kangaroo illustration.
· Eliminate the ginger snap cookies and white chocolate chips from the scene.
· Add glasses with blue-rimmed edges filled with ice and reddish-brown liquid, garnished with a slice of fruit.
· Incorporate a dried, deep red hibiscus flower on a skewer for a striking embellishment.
· Introduce a round, white ceramic bowl filled with dried flowers for visual contrast.
· Place a neat, light gray napkin beside the sangria bottle.
· Scatter dark red dried flowers around to enhance texture and align with the drink's floral theme.

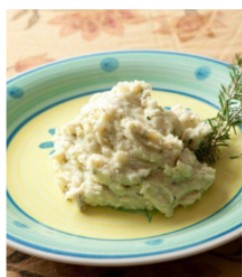
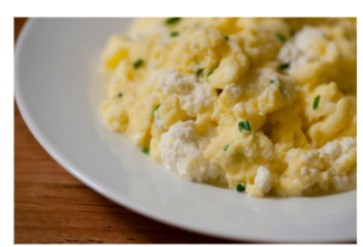

**CIRR**:
· The horn is bigger. There's a river behind the goat.

**CIRR$_R$**:
· Remove the ram and replace it with an ibex featuring large curved horns, brown fur, and a stout build with a short tail.
· Alter the posture of the animal to appear more alert.
· Transform the ground from patchy green grass to rocky terrain with light brown and gray hues.
· Incorporate scattered snow patches to the terrain for a rougher texture.
· Remove the patchy and uneven green grass from the scene.

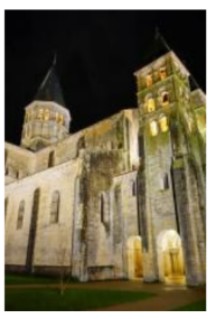
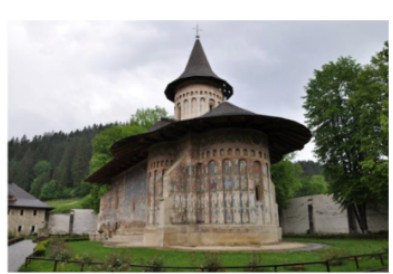

**CIRR**:
· Make the plate bigger and add more food.

**CIRR$_R$**:
· Make the plate bigger and add more food.
· Change the multicolored striped plate to a white porcelain plate with a smooth surface and shiny finish.
· Remove the rosemary sprig and add chopped chives sprinkled on top, providing a vibrant green contrast.
· Replace the floral-patterned woven tablecloth with a wooden table surface featuring a light brown color and grain patterns.

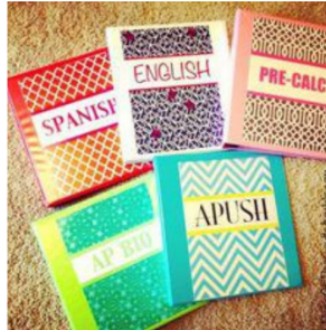
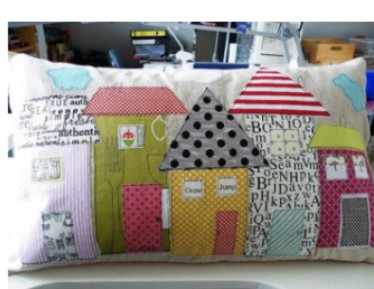

**CIRR**:
· A small construction in the middle of the moutains.

**CIRR$_R$**:
· Remove the church tower and archway for a more streamlined structure.
· Transform the stone facade and multiple spires into a single cylindrical structure with fresco-covered walls.
· Replace the angled roof with a conical roof featuring dark shingles and a wide overhang.
· Add lush green trees flanking the church and create a backdrop of rolling mountains.
· Substitute the grass lawn with a lush, natural-textured grass surrounding the structure.

**CIRR**:
· Change to a cushion showing different patterns and colours.

**CIRR$_R$**:
· Remove the five binders completely.
· Add a large rectangular pillow with a soft fabric, featuring a white background and colorful house appliqués.
· Introduce several geometric-shaped houses with distinct features such as red checkered roofs and textured beige walls and distribute them across the scene.
· Incorporate sky elements like small embroidered blue clouds with soft edges spread across the top for added decoration.
· Integrate distinct small architectural features like pink and green doors on some houses for visual variety.

Figure 1. Example original reference texts from CIRR [1] and generated text differences from the CIRR$_R$ dataset generated using our synthetic data generation pipeline.

| Query Image | Target Image | Text Modifiers |
|---|---|---|

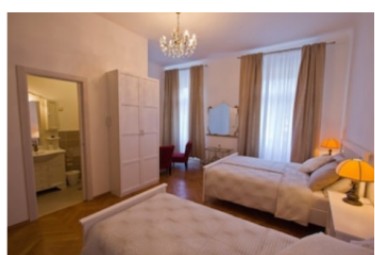
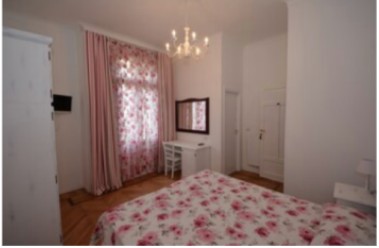

**Hotel-CIR:**
• Replace the twin beds with a queen-sized bed.
• Add a floral bedspread to match the center placement of the queen-size bed.
• Change the pillow covers to white with a floral print.
• Switch the orange lampshade to a white one.
• Replace the cone-shaped lamp shade with a cylindrical one.
• Adjust the floor length curtains to a pink color with a floral print.
• Reduce the pair of windows to a single window.
• Add a floral print blanket with soft texture and intricate pattern.
• Install a dark wooden framed mirror mounted on the wall above the desk.
• Replace the red upholstered chair with a white wooden chair.
• Add a white desk with a drawer next to the mirror.

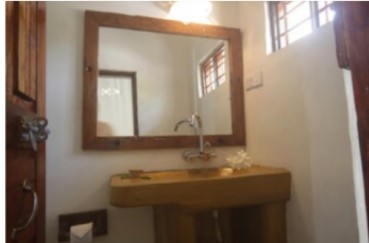
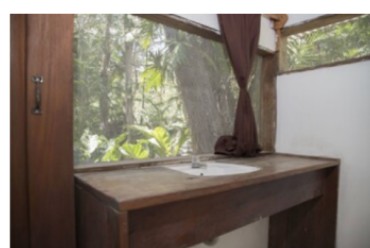

**Hotel-CIR:**
• Lower the bed closer to the ground
• Remove the visible bed frame
• Reduce the visible pillows to two per bed
• Use green decorative pillows instead of dark ones
• Place one decorative pillow centrally on each bed
• Attach the headboard to the wall instead of the bed
• Extend the headboard horizontally across the wall
• Integrate lighting fixtures into the headboard
• Introduce light yellow walls for a brighter appearance
• Ensure the window has a Roman blind with a light-colored cover
• Add a small, light-colored wooden table between the beds
• Include a piece of small, abstract artwork with a black, geometric design on the wall
• Incorporate a modern, dark-colored, geometric-shaped lighting fixture hanging from the ceiling above the foot of the first bed

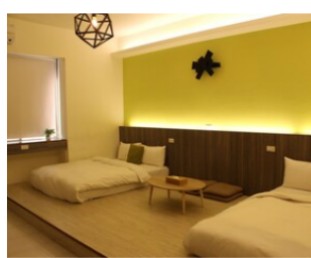
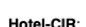

**Hotel-CIR:**
• Enlarge the window to a larger size.
• Replace the wooden frame of the window with a natural wood frame
• Move the window to be situated above the sink.
• Add a view of the outdoors outside the window.
• Change the window pane style to a single pane.
• Install brown fabric curtains hung on a wooden rod.
• Pull the curtains to one side and add a decorative tie.
• Update the countertop to have a wooden material.
• Extend the countertop to a larger surface area."

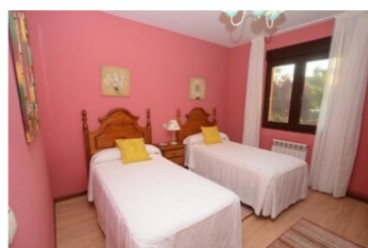
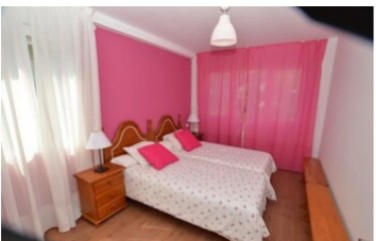

**Hotel-CIR:**
• Change the white bedding to include a small pattern
• Place the beds side by side to create a matching set
• Add an additional pink pillow to each bed
• Modify the headboard to have intricate cuts at the top
• Replace the nightstand with one that has three drawers and squared legs
• Swap out the floral bedside lamp for a modern white-colored one
• Center the large rectangular window on the wall and ensure it has a white frame
• Adjust the curtains to have a sheer white outer layer and a solid pink inner layer
• Transform one wall into a pink accent wall while keeping other walls white
• Darken the wooden flooring to a medium brown color and use hardwood material instead of laminate
• Add a light brown wooden bench against the wall for additional sitting or storage space.

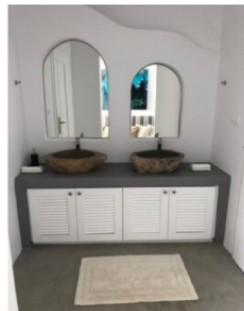
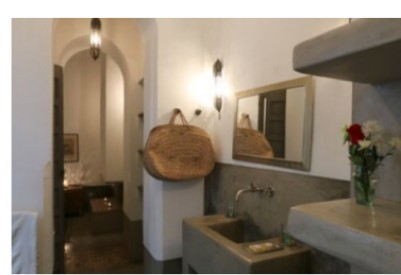

**Hotel-CIR:**
• Change the vanity mirror shape to a rectangular design.
• Add a light wooden frame to the vanity mirror.
• Reduce the vanity mirror size to medium.
• Reorient the vanity mirror to a horizontal position.
• Replace the existing sink style with a rectangular shape.
• Change the sink color to gray.
• Integrate the sink with the countertop.
• Switch to a single sink style.
• Replace the vessel sink with an under-mount sink.
• Incorporate a metal faucet into the sink design.
• Ensure the sink has a spacious basin.
• Match the countertop material to stone.
• Add a sconce light with a metal material and a dark finish.
• Mount the sconce light vertically.
• Include a flower vase with a cylindrical glass design on the countertop.
• Always have red and white flowers in the flower vase.

Figure 2. Example generated text differences for the Hotel-CIR dataset using our synthetic data generation pipeline.