# OpenReview forum: "good4cir: Generating Detailed Synthetic Captions for Composed Image Retrieval"
_thecvf.com/CVPR/2025/Workshop/SyntaGen — SyntaGen 2025 Poster_

### Official Review · Reviewer_461d · 2025-03-26

**Rating:** 6
**Confidence:** 4

**Review:**

**Summary**
The paper proposes a novel pipeline to generate high-quality synthetic annotations for datasets related to composed image retrieval (CIR). The pipeline leverages Chat GPT-4o to extract object-level descriptions, generate consistent descriptions for target images, and synthesize natural language modifications to describe the transformations. Experiments are provided to prove the effectiveness of the generation framework.

**Strength**
- The paper is well written, with clear explanation and illustration for each part of the pipeline.
- The pipeline results in more detailed instructions while maintaining the correctness of each instruction.
- Model performance increases when using the dataset collected by the proposed pipeline, especially in zero-shot benchmark, proving the effectiveness of the proposed pipeline.

**Weakness**
- There can be a lack of diversity of instructions style, causing the model to overfit to a certain style of instructions.
- Performance varies across domains; for instance, combining Hotel-CIR with CIRR_R reduces performance, indicating domain-specific training is still needed when using the released dataset.

---

### Official Review · Reviewer_HWET · 2025-03-27

**Rating:** 6
**Confidence:** 3

**Review:**

**Summary**

The paper tackles the problem of composed image retrieval, which allows a user to search images using a reference image with a text prompt describing the changes from the reference to obtain the target. In this paper the authors highlight the problem of simple text description in existing dataset, which can cause ambiguities when performing retrieval. To combat this, the paper proposed a 3 stage approach to re-generate a new text description using a Vision Language Model. Results on existing benchmarks show that the method can increase performance of existing CIR models.

**Strengths**

- The paper writing is clear and easy to understand.
- The method is straightforward, utilizing an VLM (GPT-4o) and proposing a 3 stage prompting scheme to ensure consistent text description.
- The new Hotel CIRR dataset offers a new (and more practical) evaluation setting for CIR tasks

**Weaknesses**

- Cost of annotation. While the authors note that the cost of re-annotating the CIRR dataset is $200. The use of close-sourced Vision Language Models (VLMs) is still a drawback when it comes to scaling. If the authors can provide some details regarding the use of open-source VLMs with the proposed 3 stage annotation pipeline, it would strengthen the paper.
- Results in Figure 2 is a bit confusing. Using the augmented CIRR_R dataset alone will result in significantly worse results compared to using CIRR. This shows that the repurposed text may not be as effective as the author claim. While combining CIRR + CIRR_R results in better performance, it could just be a result from the additional text pair, i.e the proposed text re-annotating framework is only an augmentation method. To strengthen the paper, the authors should also include CIRR + CIRR_R (naive one-stage approach) in Figure 2. This will show that their proposed 3 stage approach is more effective.

---

### Official Review · Reviewer_Fogq · 2025-03-27
**The paper proposes a simple yet effective pipeline for generating synthetic captions for Composed Image Retrieval with vision-language models**

**Rating:** 5
**Confidence:** 4

**Review:**

This paper addresses the limitations of prior datasets for CIR, which suffers from ambiguity and insufficient annotations by utilizing vision-language models for detailed captioning. The proposed dataset enhances retrieval accuracy, especially for fine-grained modifications.

Strengths:
- The paper is well-written and easy to understand.
- The generation pipeline is simple and straightforward, using GPT-4o for scalable and consistent annotations.

Weakness:
- Reliance on VLMs may introduce biases and hallucinations, a discussion or detailed exploration on this issue would strengthen the work.
- Additional experiments with other vision-language models are needed, as GPT-4o is closed-source and costly.

---

### Decision · Program_Chairs · 2025-03-30

**Decision:**

Accept (Poster)

**Comment:**

This paper received a Marginal Reject, a Marginal Accept, and a Marginal Accept. All reviewers agreed that the paper is well-written, and the method is simple and straightforward, yet leads to improved performance. The weaknesses raised by the first reviewer include potential biases and hallucinations inherited from VLMs and its reliance on costly, closed-sourced GPT-4o, shared also by the second reviewer. Nonetheless, the ACs agreed that, despite some limitations, the paper offers a practical and effective pipeline and recommends acceptance.

The authors are strongly encouraged to discuss the potential biases or diversity issues raised by the reviewers, expand the ablated configurations to further justify the benefits of the three-stage pipeline, and release the datasets.